# Measurement and Control System for Atomic Force Microscope Based on Quartz Tuning Fork Self-Induction Probe

**DOI:** 10.3390/mi14010227

**Published:** 2023-01-15

**Authors:** Yongzhen Luo, Xidong Ding, Tianci Chen, Tao Su, Dihu Chen

**Affiliations:** 1School of Electronics and Information Technology, Sun Yat-sen University, Guangzhou 510275, China; 2School of Physics, Sun Yat-sen University, Guangzhou 510275, China

**Keywords:** atomic force microscope, measurement and control system, embedded control, quartz tuning fork, self-inductive probe, frequency modulation

## Abstract

In this paper, we introduce a low-cost, expansible, and compatible measurement and control system for atomic force microscopes (AFM) based on a quartz tuning fork (QTF) self-sensing probe and frequency modulation, which is mainly composed of an embedded control system and a probe system. The embedded control system is based on a dual-core OMAPL138 microprocessor (DSP + ARM) equipped with 16 channels of a 16-bit high-precision general analog-to-digital converter (ADC) and a 16-bit high-precision general digital-to-analog converter (DAC), six channels of an analog-to-digital converter with a second-order anti-aliasing filter, four channels of a direct digital frequency synthesizer (DDS), a digital input and output (DIO) interface, and other peripherals. The uniqueness of the system hardware lies in the design of a high-precision and low-noise digital—analog hybrid lock-in amplifier (LIA), which is used to detect and track the frequency and phase of the QTF probe response signal. In terms of the system software, a software difference frequency detection method based on a digital signal processor (DSP) is implemented to detect the frequency change caused by the force gradient between the tip and the sample, and the relative error of frequency measurement is less than 3%. For the probe system, a self-sensing probe controller, including an automatic gain control (AGC) self-excitation circuit, is designed for a homemade balanced QTF self-sensing probe with a high quality factor (Q value) in an atmospheric environment. We measured the quality factor (Q value) of the balanced QTF self-sensing probes with different lengths of tungsten tips and successfully realized AFM topography imaging with a tungsten-tip QTF probe 3 mm in length. The results show that the QTF-based self-sensing probe and the developed AFM measurement and control system can obtain high quality surface topography scanning images in an atmospheric environment.

## 1. Introduction

Since the invention of the atomic force microscope (AFM) in 1986 [1], it has been increasingly used as a nano-scale or even an atomic-scale imaging and manipulation tool in the fields of nano-surface physics, chemistry, biology, and materials science. After undergoing nearly 30 years of development, the AFM-based scanning probe microscope (SPM) family has expanded and developed rapidly, leading to innovations such as electrostatic force microscopy (EFM), magnetic force microscopy (MFM), fluidic force microscopy (FluidFM), and piezoresponse force microscopy (PFM) [2]. Commercial AFM systems introduced in the early years are not easy to adapt to new methods and technologies. Adding electronic hardware is inconvenient, manufacturers need to provide optional hardware, and software modification is difficult [3,4,5]. With improved microprocessor performance, it is possible to use high-performance computers to process digital data and huge memory data. At present, most of the systems used to control AFMs employ commercially available computer hardware and dedicated operating system software [6]. For example, in 1989, Piner et al. designed a scanning tunneling microscope (STM) computer system to directly control the tunnel barrier width [7]. In 2008, Aloisi et al. used computers to develop and implement the functions of the AFM and realized the control of the real-time feedback loop of the scanning probe microscope (SPM). However, using a commercial computer as the AFM control system limits new technology research and performance improvement, and further improvement of the resolution or the scanning speed often requires specially customized modules [8].

Traditional desktop computers are no longer able to meet the requirements of many modern applications, which require such features as portability, low latency, parallelization, reconfigurability, networking, multi-platform compatibility, distributed processing, and low cost. In recent years, embedded systems based on microcontroller units (MCUs), digital signal processors (DSPs), or field programmable gate arrays (FPGAs) have been increasingly used in scientific instruments, especially in the AFM [9]. For example, Wong et al. proposed an STM and AFM control system with an open architecture based on AT&T’s DSP32C DSP, which not only greatly simplifies the complex AFM system but also makes it possible to achieve functions that were not easy or impossible to achieve with analog electronic equipment [10]. In 2020, Davis et al. also implemented a feedback loop for estimating the free oscillation frequency and constant frequency of a frequency-modulated non-contact AFM self-induced cantilever in an FPGA [11]. The DSP is essentially a computer dedicated to data processing. For AFM systems that need to process massive amounts of data, the DSP has more advantages than other microprocessors, such as the FPGA. However, in the AFM system, in addition to processing data, the processor resources are allocated for controlling the system, such as controlling the analog-to-digital converter (ADC), the digital-to-analog converter (DAC), the digital input and output (DIO), the stepper motor, and feedback, which consumes great amounts of DSP resources and affects efficiency. Therefore, there is an urgent need to develop higher-performance and efficient AFM control systems.

The original force sensor of the AFM is a micro-machined silicon cantilever that uses optical methods to detect its deflection, and it works in the static contact mode. However, the sensor has some shortcomings: (1) It is difficult to measure the static deflection of a small spring with a good signal-to-noise ratio; (2) contact between the tip and the sample can cause tip wear; and (3) chemical bonding forces between atoms are usually suppressed by larger van der Waals forces. These drawbacks can be addressed by ensuring that the sensor works in a dynamic mode [12], which includes two modes of operation: amplitude modulation atomic force microscopy (AM-AFM) and frequency modulation atomic force microscopy (FM-AFM). Many reports have shown that FM-AFM is usually one or several orders of magnitude more sensitive than AM-AFM [13,14]. In FM-AFM, the cantilever oscillates mechanically at its eigenfrequency $f_{0}$, detects the change in the resonant frequency (Δf) caused by force, and uses it as feedback to adjust the distance between the tip and sample [15,16]. For quartz tuning fork probes with an extremely high quality factor (Q value) and good frequency stability, FM-AFM is undoubtedly the preferred working mode [17]. Shortly after the invention of the AFM, qPlus sensors based on quartz tuning forks (QTFs) became the leading instruments for imaging at atomic and subatomic spatial resolution in ultra-high-vacuum environments [18,19,20]. In recent years, there has been increasing focus on the AFM imaging technology of the QTF self-induction probe in an atmospheric environment. For example, Akiyama et al. proposed a new probe (Akiyama Probe) based on a QTF and a U-shaped micro-machined cantilever and silicon tip and successfully imaged a single atomic step of highly oriented pyrolytic graphite (HOPG) in the indirect contact mode [21]. Li et al. developed an AFM experimental device for undergraduate teaching using a homemade QTF self-induction probe [22]. However, although their work improves the use of commercial instruments and equipment, the cost advantage is not reflected, and more importantly, the device is limited by the fact that it is used for commercial equipment, which is not conducive to the expansion of functions and the improvement of performance.

A key factor in the realization of FM-AFM technology is the detection of frequency. Several methods can be used to detect the vibration frequency of the probe, such as digital or analog frequency demodulation methods and lock-in amplifier (LIA) methods [23,24,25]. As early as 1991, Albrecht et al. used an analog phase demodulator to detect the frequency of the high-Q-value probe in FM-AFM [13]. Subsequently, Akiyama et al. used a commercial lock-in amplifier (easyPLL) to track the frequency of the AFM probe [26]. Although these methods have achieved good results in FM-AFM, digital or analog demodulators will add additional hardware circuits, complicating the system, while commercial lock-in amplifiers increase the cost due to the expensive phase-locked amplifier.

In this study, we improved on some of the above-mentioned problems, and herein we present the latest system we have developed: an AFM control system based on the OMAPL138 dual-core (DSP + ARM) microprocessor, including the control hardware and the software algorithm. First, we introduce the hardware required for the AFM function: a microprocessor and the peripherals. We discuss our low-cost and high-precision lock-in amplifier with a digital–analog hybrid structure, which helps to detect and track the frequency and phase of the FM-AFM self-induced signal. In addition, we present the fabrication of the balanced QTF probe with a high Q value and its driving control. Then, we introduce the software for our system, a software frequency measurement algorithm, and explain its measurement principle and accuracy. Finally, we present images of the surface topography of a sample obtained using the homemade QTF probe and AFM measurement and control system.

## 2. Hardware Control System

Figure 1 presents the overall schematic diagram of the AFM control system and instrument based on the quartz tuning fork (QTF) probe. It consists of two parts: the embedded control system and the probe system. The system is equipped with a host computer for data interaction with the embedded system, such as for setting scanning parameters and displaying images. Here, we first introduce the hardware structure of the embedded control system and the probe system of the AFM based on the QTF self-induction probe.

### 2.1. Embedded Control System

As shown in Figure 1, the embedded control system of the AFM is composed primarily of a microprocessor and its peripherals. The system also includes a lock-in amplifier with a digital–analog hybrid structure for frequency locking and detection, an analog high-voltage amplification module for amplifying the scanning signal, a motor controller, and a programmable gain amplifier. All the analog signals except the scanning signal and the motor control signal are processed by the programmable gain amplifier. This section focuses on microprocessors and peripherals and digital–analog hybrid lock-in amplifiers. For the control of the analog high-voltage amplifier module and the drive of the motor, we have adopted the conventional design, which we have not described in detail in this paper.

#### 2.1.1. Microprocessor and Peripherals

The core of the system is Instruments’ dual-core microprocessor OMAPL138 (DSP + ARM), the highest working frequency is 456 MHz, and the data communication between the DSP and the ARM core is via the software toolkit Syslink, provided by Texas Instruments. The embedded control system with OMAPL138 as the core is equipped with many peripheral modules necessary for the AFM control system, including multiple channels of analog-to-digital converters (ADCs), digital-to-analog converters (DACs), digital input and output interfaces (DIOs), and direct digital frequency synthesizers (DDSs). The DAC is mainly controlled by the ARM core of the microprocessor. The DAC uses TI’s 16-bit, high-precision DAC7744E with an output range of ±10V and an update rate of up to 2MSPS. Up to 24 DAC channels are designed to provide enough analog signal outputs to control the scanning signals of the piezoelectric ceramic tube (PZT) scanner in the XYZ axis and the bias voltage signals required by different modes (such as EFM), wherein the scanning signal of the Z axis is in charge of a DSP core and is output by the DAC after being calculated by a PID feedback control algorithm. After the XYZ scanning signal passes through the analog high-voltage amplifying module in the system, the obtained voltage is directly applied to the piezoelectric ceramic scanning tube to control its motion. The scanning signal is amplified by a factor of 15, the maximum output voltage is ±10 V, the amplified voltage can reach ±150 V, and the corresponding scanning range is 30μm. The DIO is also controlled by the ARM core and provided by a complex programmable logic device (CPLD), which can obtain a digital output of up to 45 channels and a digital input of up to six channels. Some of the digital output signal (DO) signals control the external CPLD, which generates stepper motor control signals, while the other DO signals are used to select the type of probe (such as an ordinary cantilever probe or a tuning fork probe), switch the control signal (such as signal switching of the AGC circuit), and control the program-controlled gain amplifier for amplifying or reducing the analog signal.

The DSP processes the data acquired by the ADC. A 16-channel general-purpose ADC also uses TI’s 16-bit, high-precision ADS8568 with an output range of ±10V and a maximum read speed of 510 kSPS. In addition, there are six-channel synchronous sampling ADCs with second-order anti-aliasing analog filters using AD7606 by Analog Devices, which has an output range of ±10V and a reading speed of 200 kSPS. The signal read by the ADC is mainly the response signal of the probe, which reflects the change in the probe’s eigenfrequency when the probe is affected by the force gradient. The frequency and phase responses are detected and tracked by the dedicated high-precision digital–analog hybrid lock-in amplifier in the system and used as the input of the PID feedback controller for imaging and measurement in various modes. The system provides 22 ADC channels to meet the signal acquisition needs of different functions. In addition, the DSP core controls the output signals of the direct digital synthesizer (DDS), which are used as the excitation signals of the probe and the reference signals of the lock-in amplifier. Generating stable frequency and phase signals in this system are AD9831 DDS devices from Analog Devices, Inc. These are numerically controlled oscillators that can support clock rates up to 25 MHz with a frequency control accuracy of 1 part in 4 billion. They can completely meet the signal requirements for the excitation of the QTF probe and the reference input of the lock-in amplifier in the system.

In the AFM imaging system of the QTF probe, the DAC and the ADC play vital roles. In the following section, we describe testing the functionality of the DAC and the ADC separately.

The ADC ADS8568 is controlled by the DSP to collect 1000 sine wave signals with a frequency of 400 Hz and a peak-to-peak value of 4.0 V and 1000 sine wave signals with a frequency of 2 kHz and a peak-to-peak value of 20.0 V. The oscilloscope waveform of the input signal was compared with the data collected by the ADC, as shown in Figure 2. The data acquisition function of the ADC is normal, and the ADC acquisition speed is about 5 μs.

Since the ADC acquires the imaging signal, the resolution of the image is closely related to the signal-to-noise ratio of the ADC channel, and we evaluated the noise levels of the 16 common ADC channels in the system. Figure 3 shows the noise level of channel 0 of the ADC (ADC0). Figure 3a shows the waveform of the noise. The root-mean-square (RMS) value of the noise is about 2.66 mV. Figure 3b shows the distribution interval of the noise, which follows statistical laws, and the distribution on the vertical axis basically satisfies a normal distribution. The range of ±3.3σ in the normal distribution can contain 99.9% occurrence probability, and the peak-to-peak value of the maximum noise is about 6.6 times that of its effective value.

Table 1 presents the RMS noise of the tested 16-channel general-purpose ADCs and shows that when the peak-to-peak signal is 10 V, the signal-to-noise ratio (SNR) of the noise data acquisition system can reach a value of more than 85.5 dB; that is, the significance bit of the ADC can reach a value of 15 bits, which can fully meet the requirements of atomic-level-resolution imaging of the AFM system.

For the system to achieve atom-resolved (~ 0.1 nm) images, the minimum distance required for the ADC to recognize the signal is about 0.01 nm, and the corresponding voltage signal is about 0.0067 mV. The resolution of the 16-bit ADC with an output range of ±10 V is 0.3 mV, so the tuning fork controller or the program-controlled amplifier of the system usually amplifies the self-induction signal by at least two orders of magnitude. The signal amplified by the hardware also helps to improve the signal-to-noise ratio of the system.

The method for testing the DAC is the reverse of the method for testing the ADC. The DSP controls the DAC, because of which the DAC outputs a triangular wave and a sawtooth wave with fixed frequencies and amplitudes. The output signal is connected to the oscilloscope, and the result is shown in Figure 4. It can be seen that the DAC is functioning properly.

#### 2.1.2. Digital–Analog Hybrid Lock-In Amplifier

The lock-in amplifier is used to detect a weak signal. It uses a reference signal with the same frequency as the measured signal as a basis for comparison and responds to only those signal components with the same frequency (or frequency doubling) and phase as the reference signal in the measured signal. It can greatly suppress the noise and improve the signal-to-noise ratio. In addition, the lock-in amplifier has high detection sensitivity, so it is suitable for detecting the signal of the frequency modulation mode of the AFM control system based on the QTF probe.

At present, there are a variety of commercial phase-locked instruments to choose from. However, these have some problems. For example, they are unable to adjust the phase-locked frequency in time through the digital control system, and they are costly. In addition, the direct application of the digital phase-locked algorithm puts a great burden on the digital signal processing system. To solve these problems, in our system, we designed a lock-in amplifier with a digital–analog hybrid structure (Figure 5) that is specially used to detect and track the FM-AFM signal. The analog part of the lock-in amplifier was designed by Analog Devices, and the digital part was completed by the DSP algorithm.

In FM-AFM, the tuning fork probe is self-excited under the control of the AGC loop and vibrates freely at an eigenfrequency of $\omega_{0}$. When there is a force gradient between the tip and the sample, the eigenfrequency of the tuning fork will be shifted to $\omega_{1}$. The trans-impedance amplifier (TIA) circuit in the tuning fork controller amplifies the induced current of the tuning fork probe so that the signal to be measured is obtained:(1)$$h(t)=A\sin(\omega_{1}t+\varphi_{1})$$

After entering the lock-in amplifier, the signal to be measured is filtered by a 50/100 Hz notch filter and a self-tracking narrow-band filter with noise attenuation not less than 40 dB in the channel to be measured. The reference channel signals are two orthogonal reference signals generated by the DDS that are the same as the probe eigenfrequency $\omega_{0}$ ($\omega_{0}=2\pi f_{0}$). They are also simultaneously controlled by the feedback module in the digital arithmetic unit, and the phases of the reference signals are adjusted in real-time to improve the resolution of phase detection. The two quadrature signals are
(2)$$f(t)=B\sin(\omega_{0}t+\varphi_{0}),g(t)=B\cos(\omega_{0}t+\varphi_{0})$$
and they are each multiplied by the filtered signal of the channel in order to be measured by a four-quadrant analog multiplier, and the following equations are obtained:(3)$$h_{1}(t)=f(t)\times h(t)=\frac{AB}{2}\{\cos[(\omega_{1}-\omega_{0})t+(\varphi_{1}-\varphi_{0})]-\cos[(\omega_{1}+\omega_{0})t+(\varphi_{1}+\varphi_{0})]\}$$
(4)$$h_{2}(t)=f(t)\times h(t)=\frac{AB}{2}\{\sin[(\omega_{1}-\omega_{0})t+(\varphi_{1}-\varphi_{0})]-\sin[(\omega_{1}+\omega_{0})t+(\varphi_{1}+\varphi_{0})]\}$$

The output is filtered for the first time by an analog low-pass filter (ALPF), which in our system is a second-order active low-pass filter composed of an operational amplifier with a bandwidth of 30–50 kHz that forms a cascade filter with the digital low-pass filter (DLPF) of the digital part. The analog low-pass filter does not need to completely remove the interference signal, which allows the existence of the mixed signal in the passband, and simultaneously limits the maximum frequency of the signal that needs to be digitized to less than half of the sampling frequency without distortion, to avoid false signals in the digital signal.

The ADC with a second-order anti-aliasing analog filter converts the above two analog signals into digital signals, which enter the DSP for digital processing; high-frequency interference signals and aliasing signals are filtered by a digital filter for the second time and the two paths of signals $h_{1}\left(t\right)$ and $h_{2}\left(t\right)$ are obtained after being controlled and adjusted by a digital arithmetic unit. They are
(5)$$h_{1}(t)=\frac{\mathrm{AB}}{2}\cos[(\omega_{1}-\omega_{0})t+(\varphi_{1}-\varphi_{0})]$$
(6)$$h_{2}(t)=\frac{\mathrm{AB}}{2}\sin[(\omega_{1}-\omega_{0})t+(\varphi_{1}-\varphi_{0})]$$
where A is the amplitude of the phase-locked signal, B is the amplitude of the reference signal,$\omega_{0}$ and $\omega_{1}$ are the frequencies of the reference signal and the signal to be measured, respectively, and $\varphi_{0}$ and $\varphi_{1}$ are the phases of the reference signal and the signal to be measured, respectively. The result after the digital operation is the final output of the digital–analog hybrid lock-in amplifier, and the result can be calculated by our software frequency measurement algorithm.

The hybrid structure lock-in amplifier can improve the signal-to-noise ratio well through the analog and digital hybrid cascade filter and the digital control self-tracking narrow-band filter. In FM-AFM, using the analog part with the analog multiplier and the digital part with the digital signal processor as the cores and improving the algorithm, we can realize the high-precision detection of a frequency signal in the system.

### 2.2. Probe System

AFM’s probe system consists of (1) a probe base that includes a scanner and a stepper motor and (2) a tuning fork pre-controller that includes probe type selection and QTF probe drive modules. In this section, we describe the characteristics of the homemade balanced QTF probe with a high Q value and the tuning fork pre-controller used for the QTF probe.

#### 2.2.1. Characteristics of the QTF Probe

For FM-AFM, the quality factor (Q value) of the probe is a key parameter closely related to sensitivity. The image resolution of a probe with a high Q value working in FM-AFM mode is one or even several orders of magnitude higher than that in AM-AFM mode. Thanks to the resonant characteristics of the QTF, one of the most important characteristics of the QTF probe is its high Q value, which can be as high as more than 10,000 in an atmospheric environment without an additional tip, so the probe is suitable for fabricating AFM force sensors. Figure 6 shows a comparison of the Q values between the QTF probe and an ordinary cantilever beam probe (the Q value of the balanced QTF probe is about 3000, while that of the cantilever probe is about 350). The abscissa in Figure 6 represents the difference between the excitation signal frequency and the probe eigenfrequency (Δf). It can be seen from Figure 6 that the QTF probe has a higher Q value than the ordinary cantilever beam probe and is, therefore, more suitable for measurement imaging in FM-AFM mode.

When fabricating AFM probes with a QTF, tips made of materials such as silicon or tungsten are usually bonded or glued to the tuning fork arm, because of which the symmetry of the two fork arms is destroyed, and the Q value decreases. To maintain a high Q value of the tuning fork, we improved the balanced tuning fork probe structure proposed by Zhang et al. [27], which preserves the symmetry of the two arms of the tuning fork by employing a particular technique to bond the tip in an atmospheric environment. We control the balance of the QTF by bonding tungsten tips of the same length to the two arms of the QTF and maintaining the adhesive mass of the two arms. Figure 7 shows the fabricated QTF probe and its tip. The fabrication method is as follows: (1) Peel off the shell of the quartz crystal oscillator commonly used in quartz watches to obtain a quartz tuning fork; (2) bond a tungsten wire with a diameter of about 0.1 mm onto the two arms of the tuning fork with an adhesive; and (3) obtain the tungsten tip of the probe by electrochemical corrosion, where the curvature radii of the tungsten tip R and cone angle θ are controlled by an electrochemical corrosion circuit and the length of the probe is controlled by micrometer calipers.

The balanced QTF probe with tungsten wires bonded at both ends of the quartz tuning fork in an atmospheric environment can still maintain a Q value of more than 3000, up to about 4000, even if the resonance frequency is reduced (usually 24–26 kHz). The Q value is related to the length of the tungsten tip and the additional mass. Under the same bonding process, the longer the length of the tungsten tip, the lower the value of Q. Figure 8 shows the Q values for bonding tungsten tips of different lengths. The Q value is about 3100 when the length of the tungsten tip is 1 mm and about 1280 when it is 3 mm.

The Q value range of the tungsten-tip QTF probe prepared by the above method is generally between 1000 and 4000, and the Q value of the lighter silicon tip or the micro-machined tip can even reach 10,000. At present, there is no instrument or controller suitable for a QTF probe with a Q value in the range of 1000~10,000 on the market. Therefore, we developed a tuning fork controller suitable for this Q value range for our homemade AFM control system to drive the QTF probe in an atmospheric environment.

#### 2.2.2. Tuning Fork Pre-Circuit and Control

The pre-circuit of the tuning fork probe consists of a QTF probe-driving circuit and a self-excitation and self-induction circuit, which is a closed circuit in the AFM scanning imaging state.

Figure 9 depicts the self-excitation driving circuit of the self-sensing QTF probe of the tuning fork controller, which is mainly composed of a tuning fork probe-driving circuit and a self-excitation circuit. The tuning fork probe-driving circuit comprises a pre-stage attenuation circuit G1 (attenuation coefficient), a capacitance compensation circuit G2, and a trans-impedance amplification circuit (TIA) G3. The self-excitation circuit comprises an automatic gain controller (AGC) based on amplitude detection and a phase adjustment module. When FM-AFM works, the DO is connected to the output of the self-excitation circuit as the switch control S1, and then the excitation signal is applied to the excitation end of the probe through the pre-stage attenuation circuit. The attenuation coefficient of the pre-stage attenuation circuit is also adjustable via digital signal control. G2 is used to eliminate the effect of the parasitic capacitance of the quartz tuning fork. After the response signal of the probe passes through the TIA circuit (G3), the current signal is converted into voltage and output to the digital–analog hybrid lock-in amplifier and is also input to the self-excitation circuit. The amplitude and phase of the signal are automatically adjusted by the AGC, thus forming a self-excitation and self-induction closed-loop circuit system. When measuring the Q value curve, the DO controls the S1 switch to f0, a sweeping frequency signal output by the DDS, which drives the QTF probe to excite its mechanical vibration.

In addition to the homemade QTF self-sensing probe, the tuning fork control system is compatible with different types of QTF self-sensing probes, such as the commercial Akiyama Probe, and can select the type and switch the signal through the DO.

## 3. Control and Algorithm

### 3.1. Software Structure of the Embedded Control System

To meet the functional requirements of FM-AFM, we adopted a hierarchical and modular software design method to program the FM-AFM software system. The embedded control software can be divided into an application layer, a function layer, and a driver layer, as shown in Figure 10.

### 3.2. Software Frequency Measurement Algorithm

In this paper, we mainly focus on the software frequency measurement algorithm in the embedded control software and have not discussed the software implementation of the driver layer and the function layer in detail.

When the AFM measurement and control system works in the FM-AFM mode, the force gradient of the QTF probe leads to a change in its eigenfrequency (assuming that the change is Δf). By detecting the change in frequency Δf, the force gradient of the probe can be obtained. At the same time, Δf is used as the input of the PID feedback controller in the system to control the output of the *Z*-axis voltage so as to adjust the distance between the tip and the sample to obtain the topography information of the sample surface. To detect the frequency variation Δf, we propose a software algorithm based on the DSP, namely the software frequency measurement method.

After the digital–analog hybrid lock-in amplifier detects and tracks the response signal of the QTF probe, the real-time frequency and phase information of the mechanical vibration of the QTF probe can be obtained. The ADC collects the information and transmits it in the form of a data stream to the DSP, which calculates and processes the information. Assuming $t=t_{1}$, the DSP acquires signals from two analog multipliers:(7)$$H_{1}=M\cos(\omega_{m}t_{1}+\varphi_{m}),H_{2}=M\sin(\omega_{m}t_{1}+\varphi_{m})$$
where $\omega_{m}=\omega_{1}-\omega_{2},\varphi_{m}=\varphi_{1}-\varphi_{2},M=\frac{AB}{2}.$

The DSP first performs the division operation and then calculates the arctangent value using the following equations:(8)$$H_{0}=\frac{H_{2}}{H_{1}}=\tan(\omega_{m}t_{1}+\varphi_{m})$$
(9)$$\Phi_{0}=\arctan(H_{0})=\omega_{m}t_{1}+\varphi_{m}\;$$

After a fixed time interval τ, the DSP repeats the above process to obtain the phase at this time:(10)$$\Phi_{1}=\omega_{m}(t_{1}+\tau )+\varphi_{m}$$

The DSP subtracts $\Phi_{0}\;\mathrm{and}\;\Phi_{1}$ to obtain
(11)$$\Phi =\Phi_{1}-\Phi_{2}=\omega_{m}\tau$$

Since ω_0_ and τ are known quantities, $\omega_{m}$ (the variation in the probe angular frequency) can be calculated as
(12)$$\omega_{m}=\frac{\Phi }{\tau }$$
(13)$$\Delta f=\frac{\omega_{m}}{2\pi }$$

The sampling frequency of the ADC in this system is set to 125 kHz; that is, the period T = 8 μs. In general, we set the fixed time interval τ to be a multiple of the sampling period as a time window, for example, τ = 40 T or τ = 50 T, and the Δf obtained is the average amount of frequency variation within this window.

### 3.3. Relative Error of Frequency Measurement

To evaluate the accuracy of software frequency measurement, we use a signal generator to simulate the signal to be measured, and the reference signal and the frequency difference between the signal to be measured and the reference signal is Δf. The software sets the corresponding relationship between the frequency difference f and the voltage to be output, and the voltage is output through the DAC. The relative error of frequency measurement can be obtained by measuring the range of output voltage corresponding to different values of Δf. Table 2 presents the relative error of frequency measurement obtained in different time windows and shows that the length of the time window has little effect on the relative error of frequency measurement. When the reference frequency is 28 kHz, the correspondence between the frequency shift Δf and the output voltage set by the software is 1 Hz = 20 mV.

In FM-AFM, the frequency shift Δf due to a force gradient is generally given by $\Delta f\approx \frac{1}{2}\frac{f}{k}\frac{\partial F}{\partial z}$ [28], where *f* is the resonance frequency (~25 kHz herein) and *k* is the spring constant (~1000 N/m herein). Therefore, with our typical experimental conditions, such that the frequency resolution is 0.025 Hz, and the vibration amplitude is about 1 nm, the minimum measurable force sensibility is about 2 pN.

## 4. Characterization and Results

In the FM-AFM mode of the AFM measurement and control system, the frequency characteristic of the balanced QTF probe, namely the Q curve, should be measured before the image is scanned. At this time, we need to switch the S1 of the tuning fork controller in Figure 9 to external excitation mode via the digital output control; that is, the excitation signal is provided by the DDS and swept to obtain the frequency response curve. After the frequency sweep, the tuning fork controller automatically switches to the AGC self-excitation loop control mode, the probe performs self-excitation and self-induction, and the QTF probe vibrates freely at its eigenfrequency.

Figure 11 shows scanned images of a DVD disc obtained with a balanced QTF probe with a tungsten tip 3 mm in length in FM-AFM mode. Several different images were obtained by changing the scanning range while keeping the scanning speed and the image center position fixed. The scanning speed was set to 1 line/s; that is, the round-trip scanning time was 1 s. The number of scanning points was set to 800 × 800, the frequency shift reference was set to 5 Hz, and the scanning range increased from 2 μm to 8 μm. The only processing applied to the scanned image was straightening processing which flattened the image in the horizontal direction.

Figure 12 shows topographic profiles of the scan lines (y = 0) in the scanned images in Figure 11, with one scan line per scanned image. It can be seen from the scanning profile that with the same scanning speed, the smaller the scanning range, the more stable the profile, and the closer the image is to the real surface topography. For the 2 μm × 2 μm scanning range shown in Figure 12a, the sample track spacing of the DVD disc is about 750 nm, and the groove depth is about 92 nm, which is basically consistent with the standard parameters of the DVD (the track spacing is ~740 nm and the groove depth is ~104 nm).

In profile scanning, it is easy to observe the nonlinear phenomenon of the obtained pattern due to the hysteresis of the piezoelectric scanning tube; for example, the image on the left is larger than the one on the right. This nonlinear error can be improved by the hysteresis compensation of the feedback controller, which is also work we are carrying out.

## 5. Conclusions

In this study, we developed a scalable and compatible measurement and control system for an atomic force microscope (AFM) based on a quartz tuning fork (QTF) self-induction probe and frequency modulation. The system is based on the dual-core OMAPL138 microprocessor, equipped with a sufficient number of peripherals and interfaces, which has all the necessary attributes for AFM operation and realizes all the measurement and imaging functions of the AFM based on the QTF self-induction probe. A high-precision and low-noise lock-in amplifier was designed to detect and track the frequency and phase of the frequency modulation response signal of the QTF probe. A method based on a software algorithm was proposed to calculate the change in the signal frequency, and the relative error of frequency measurement is up to or better than 3% in the frequency range used. A self-sensing probe controller, including an automatic gain control (AGC) self-excitation circuit, was designed for a high Q value and a balanced QTF self-sensing probe in an atmospheric environment. We measured the Q values of balanced QTF self-induction probes with tungsten tips of different lengths and obtained surface topography images of DVD disc samples using a tungsten-tip probe 3 mm in length. The results show that high quality surface topography images can be obtained using a QTF self-induction probe and the developed AFM measurement and control system in an atmospheric environment. In future work, we plan to address some limitations of the current system, including the hysteresis of the piezoelectric scanning tube. In addition, we plan to apply the measurement and control system to other types of scanning probe microscopes (SPMs), such as the electrostatic force microscope (EFM), the scanning capacitance microscope (SCM), and the magnetic force microscope (MFM) and focus on providing novel instruments and methods for inventing new optoelectronic devices or integrated circuits.

## Figures and Tables

**Figure 1 micromachines-14-00227-f001:**
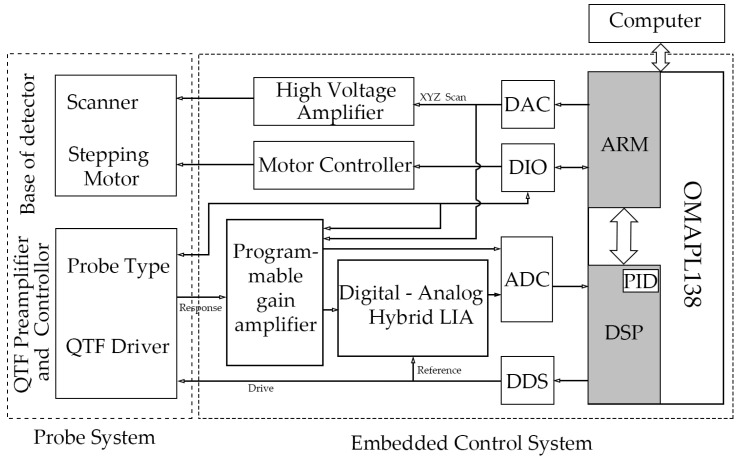
Schematic diagram of the AFM control system based on the QTF self-induction probe and the whole instrument.

**Figure 2 micromachines-14-00227-f002:**
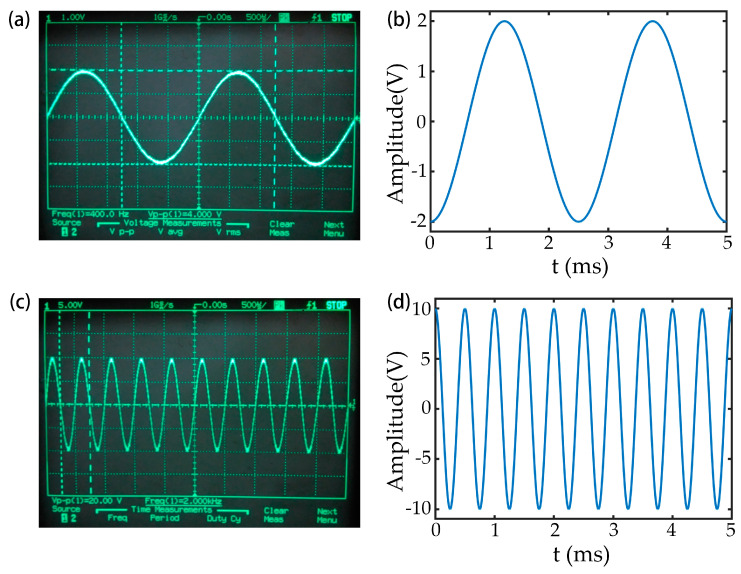
Oscilloscope waveforms of the input signal and the digital waveform collected by the AD converter controlled by the DSP: (**a**) Oscilloscope waveform with an input frequency of 400 Hz and a peak-to-peak value of 4.0 V; (**b**) waveform corresponding to Figure 2a acquired by the ADC; (**c**) oscilloscope waveform with an input frequency of 2 kHz and a peak-to-peak value of 20.0 V; (**d**) waveform corresponding to Figure 2c acquired by the ADC.

**Figure 3 micromachines-14-00227-f003:**
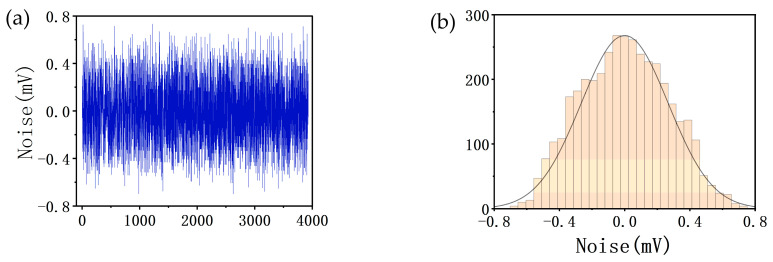
Noise of ADC0: (**a**) Noise waveform (the *x*-axis represents the collection of 4000 data points) and (**b**) noise distribution interval (the *y*-axis represents the number of data points distributed at different intervals).

**Figure 4 micromachines-14-00227-f004:**
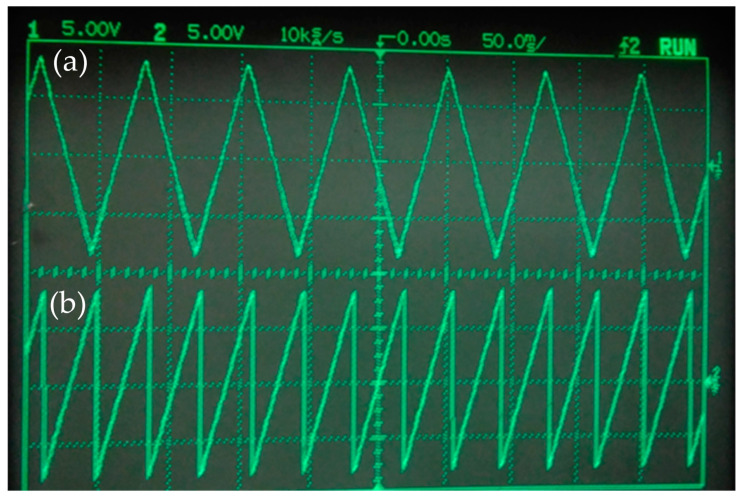
Waveforms of the DAC output signal: (**a**) triangular wave and (**b**) sawtooth wave.

**Figure 5 micromachines-14-00227-f005:**
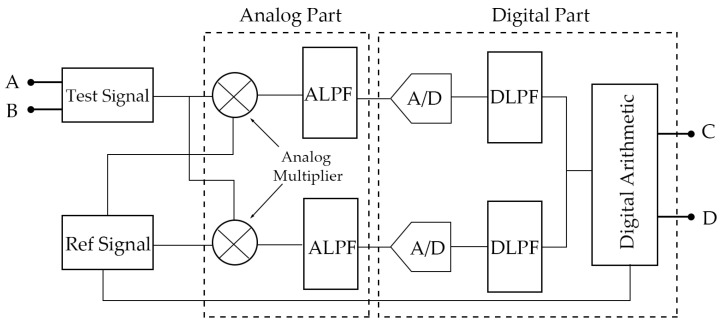
Schematic diagram of a lock-in amplifier with a digital–analog hybrid structure.

**Figure 6 micromachines-14-00227-f006:**
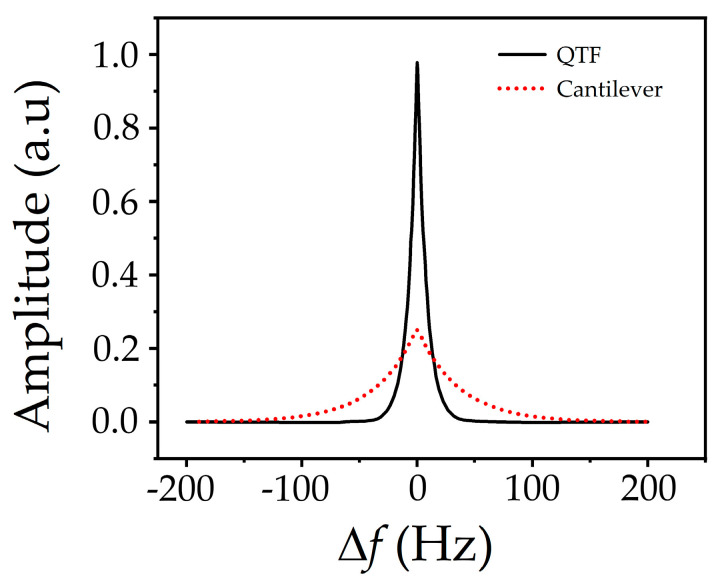
Q values of the QTF probe and the cantilever beam probe.

**Figure 7 micromachines-14-00227-f007:**
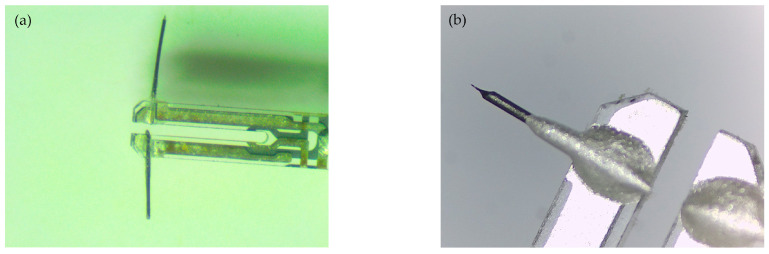
Photos of a quartz tuning fork probe: (**a**) Quartz tuning fork arm for bonding tungsten wire; (**b**) probe tip after electrochemical corrosion.

**Figure 8 micromachines-14-00227-f008:**
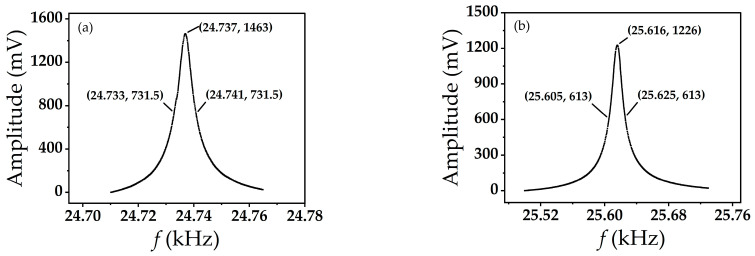
Q values of a balanced tuning fork probe: (**a**) Q value when the length of the tungsten tip is 1 mm; (**b**) Q value when the length of the tungsten tip is 3 mm.

**Figure 9 micromachines-14-00227-f009:**
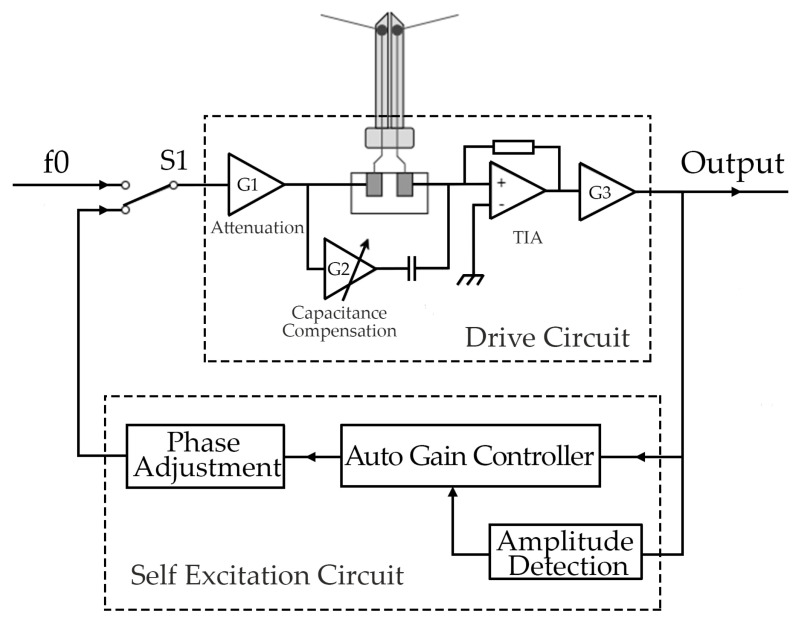
Schematic diagram of the self-excitation drive of the QTF self-induction probe.

**Figure 10 micromachines-14-00227-f010:**
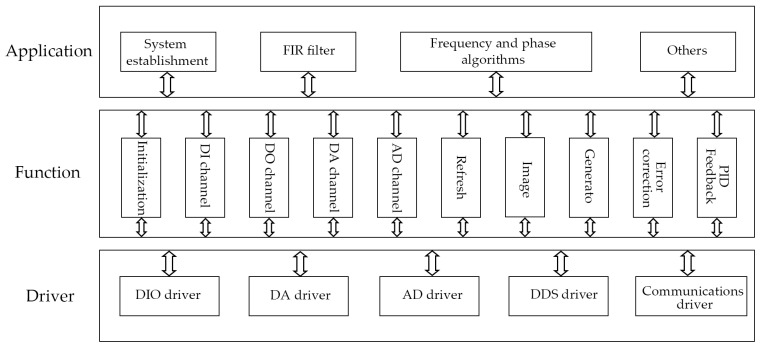
Schematic diagram of the embedded control software structure.

**Figure 11 micromachines-14-00227-f011:**
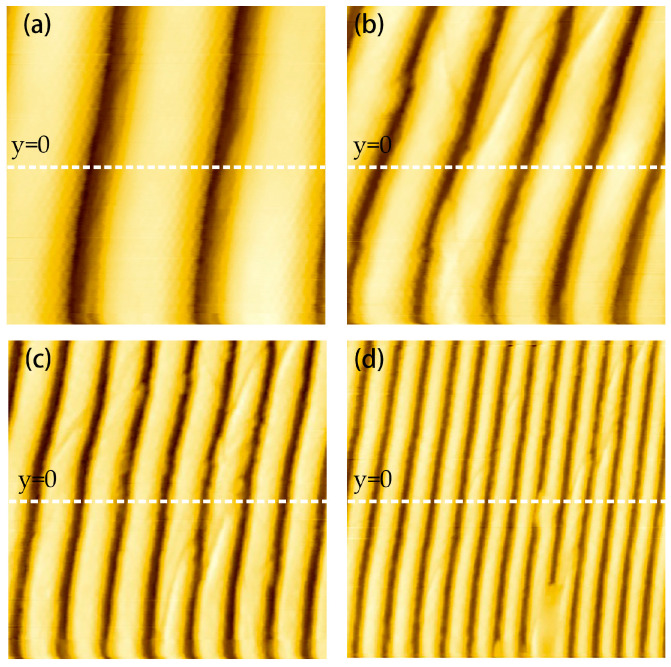
Scanned images of a DVD disc sample: (**a**) Scan range 2 μm × 2 μm; (**b**) scan range 3.3 μm × 3.3 μm; (**c**) scan range 5.3 μm × 5.3 μm; (**d**) scan range 8 × 8 μm.

**Figure 12 micromachines-14-00227-f012:**
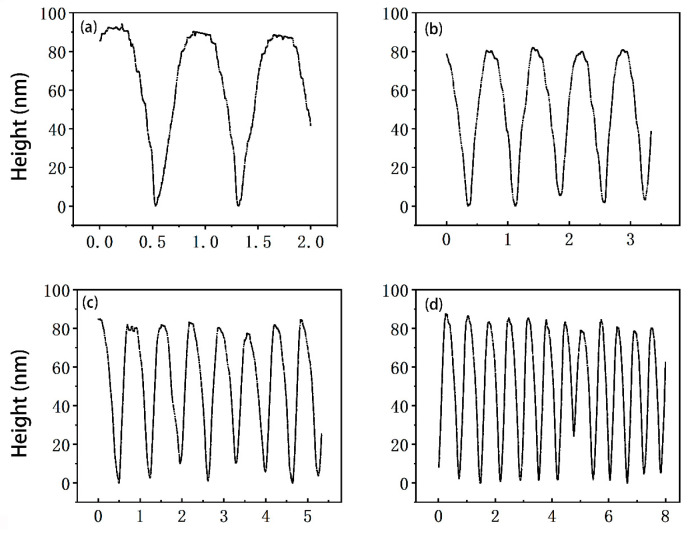
Profiles of the scanned images in Figure 11 (marked in Figure 11): (**a**) Scan range 2 μm × 2 μm; (**b**) scan range 3.3 μm × 3.3 μm; (**c**) scan range 5.3 μm × 5.3 μm; (**d**) scan range 8 μm × 8 μm.

**Table 1 micromachines-14-00227-t001:** Noise evaluation of each channel of the ADC.

**Channel**	AD0	AD1	AD2	AD3	AD4	AD5	AD6	AD7
**Noise (mV)**	0.266	0.273	0.136	0.182	0.500	0.530	0.318	0.273
**Channel**	AD8	AD9	AD10	AD11	AD12	AD13	AD14	AD15
**Noise (mV)**	0.182	0.227	0.136	0.091	0.227	0.182	0.500	0.500

**Table 2 micromachines-14-00227-t002:** Relative error of frequency measurement in different time windows.

Frequency Shift	*τ =* 40 T		*τ =* 50 T	
Δf(Hz)	Output Voltage (mV)	Frequency (Hz)	Output Voltage (mV)	Frequency (Hz)
10	190.0 ± 10	9.50 ± 0.50	192.5 ± 12.5	9.63 ± 0.63
20	392.5 ± 17.5	19.63 ± 0.88	392.5 ± 17.5	19.63 ± 0.88
30	585.0 ± 25	29.25 ± 1.25	592.5 ± 27.5	29.63 ± 1.38
40	785.0 ± 35	39.25 ± 1.75	790 ± 30	39.50 ± 1.50
50	980.0 ± 40	49.0 ± 2.00	990 ± 40	49.50 ± 2.00
60	1170.0 ± 40	58.50 ± 2.00	1185 ± 45	59.25 ± 2.25
70	1380.0 ± 50	69.00 ± 2.50	1385 ± 55	69.50 ± 2.75
80	1580.0 ± 60	79.00 ± 3.00	1580 ± 60	79.00 ± 3.00
90	1775.0 ± 65	88.75 ± 3.25	1780 ± 65	89.00 ± 3.25
100	1970.0 ± 70	98.50 ± 3.50	1980 ± 70	99.00 ± 3.50

## Data Availability

Not applicable.

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
