# Peer review of "Measurement and Control System for Atomic Force Microscope Based on Quartz Tuning Fork Self-Induction Probe"

_micromachines, 2023, doi:10.3390/mi14010227_

Round 1
Reviewer 1 Report
The authors reported on the design and demonstration of the embedded hardware for the QTF-based AFM system. In particular, for the probe system, a self-sensing probe controller was designed for a home-made balanced QTF self-sensing probe with a high quality factor in an atmospheric environment and realized high quality surface topography scanning imaging. i
I recommend for minor revision before publication. I suggest the authors discuss the force measurement capability including the minimum force sensibility resulting from the demonstration of the embedded hardware system.
Reviewer 2 Report
This manuscript presents a measurement and control system for atomic force microscope based on quartz tuning fork probe, which is mainly composed of an embedded control system and a probe system.
1. Some figures can be improved. For example, Figure 2(b) and 2(d) lack description of horizontal and vertical coordinates. Figure 3(a) lacks description of horizontal coordinate, and Figure 3(b) lacks description of vertical coordinate. Moreover, the quality of figures can be improved.
2. The authors claim that they propose a balanced tuning fork probe structure that maintains the symmetry of the two arms of the tuning fork, by bonding tungsten tips of the same length. However, as far as I know, this method has been already proposed, which was mentioned in literature “A High-Q AFM Sensor Using a Balanced Trolling Quartz Tuning Fork in the Liquid”, doi:10.3390/s18051628. So it would be better to clarify where this idea came from, and cite this reference.
3. Figure 12 shows topographic profiles of the scan lines in the scanned images in Figure 11. It would be better to sign the corresponding scan lines in Figure 11.
4. The image resolution of a probe with a high Q value working in FM-AFM mode is higher than that in AM-AFM mode. It would be better to illustrate how much the frequency resolution is in the proposed system for FM-AFM.
5. The abbreviations can be improved. For example, the abbreviation ‘DO’ first appears in Line 165, while it full name ‘digital output signal’ appears in Line 366.
